# A Dated Phylogeny of the Pantropical Genus *Dalbergia* L.f. (Leguminosae: Papilionoideae) and Its Implications for Historical Biogeography

Fabien Robert Rahaingoson [1,2] , Oyetola Oyebanji [1,2], Gregory W. Stull [1], Rong Zhang [1,*] and Ting-Shuang Yi [1,*]

1. Germplasm Bank of Wild Species, Kunming Institute of Botany, Chinese Academy of Sciences, Kunming 650201, China; fabienrobertrahaingo@mail.kib.ac.cn (F.R.R.); oyetola@mail.kib.ac.cn (O.O.); gwstull@gmail.com (G.W.S.)
2. University of Chinese Academy of Sciences, Beijing 100049, China
* Correspondence: zhangronga@mail.kib.ac.cn (R.Z.); tingshuangyi@mail.kib.ac.cn (T.-S.Y.)

**Abstract:** The genus *Dalbergia* has a pantropical distribution and comprises approximately 250 species. Previous phylogenetic studies on the genus revealed that *Dalbergia* is monophyletic and is sister to *Machaerium* and *Aeschynomene* sect. *Ochopodium*. However, due to limited samples or DNA regions in these studies, relationships among the major clades are still unresolved, and divergence dates and biogeographical history of the genus have not been addressed. In this study, phylogenetic analyses of *Dalbergia* were conducted using broad taxon sampling and a combined dataset of two plastid DNA markers (*mat*K and *rbc*L) and one nuclear marker (ITS). We evaluated the infrageneric classification of the genus based on the reconstructed tree, and investigated biogeographical history of this genus through molecular dating and ancestral area reconstruction analyses. The monophyly of *Dalbergia* was strongly supported and the genus was resolved into five major clades with high support, several of which correspond to the previous recognized sections. We inferred that *Dalbergia* originated in South America during the Early Miocene (*c*. 22.9 Ma) and achieved its current pantropical distribution through multiple recent transoceanic long-distance dispersals (LDD). We highlighted the important historical events which may explain the pantropical distribution pattern of *Dalbergia*.

**Keywords:** *Dalbergia*; Early Miocene; ITS; long-distance dispersal; *mat*K; monophyletic; *rbc*L

## 1. Introduction

The pantropical genus *Dalbergia* L.f. includes *c*. 250 species with centers of diversity in Central and South America, Africa, Madagascar, and Asia [1]. *Dalbergia* species grow in diverse habitats including tropical rain forests, dry forests, savannas, costal dunes, and rocky outcrops [2–4]. The *Dalbergia* species display a high diversity of life forms, including trees, shrubs, and woody lianas. The genus is economically important for its high-quality timber known as rosewood (e.g., Brazilian species *D. nigra* (Vell.) Allemão ex Benth., Chinese species *D. odorifera* T.C.Chen, Madagasar species *D. baronii* Baker, and Thailand species *D. cochinchinensis* Pierre), and blackwood (e.g., *D. melanoxylon* Guill. & Perr.), which are used for construction works, fine furniture and musical instruments [5]. Many *Dalbergia* species have also been used in traditional medicine and phytochemical studies [6]. Overexploitation, illegal logging, and habitat fragmentation have severely decreased the population sizes of many *Dalbergia* species.

Studies on the intrageneric classification of *Dalbergia* are limited. Only Bentham (1860) [7] carried out large-scale taxonomic study of the genus, and divided the 64 *Dalbergia* species known at that time into six series. Later, this work was followed by subsequent regional monographs based exclusively on morphological characteristics. In the Neotropics, the Brazilian species of *Dalbergia* were studied by Carvalho (1997) [2]. He divided the 41 Brazilian *Dalbergia* species into five sections based on inflorescence and fruit types.

Prain (1904) [8] classified the 86 South East Asian *Dalbergia* species into two subgenera, five sections and 24 series. Thothathri (1987) [9] categorized the 46 *Dalbergia* species present in the Indian subcontinent into four sections and seven series, based on androecium and fruit types.

Only a few molecular phylogenetic studies have been carried out to address the phylogenetic position of *Dalbergia* and its intrageneric relationships. Phylogenetic analysis of the tribe Dalbergieae, based on molecular and morphological data, placed *Dalbergia* within the Dalbergioid clade [10]. Using plastid *trn*L and *mat*K, and nuclear ribosomal ITS sequences data, *Dalbergia* were observed to be monophyletic and sister to a clade comprising the genus *Machaerium* Pers. and *Aeschynomene* sect. *Ochopodium* [11,12]. The above-mentioned studies have highlighted the relationships between *Dalbergia* and its relatives and included a limited sampling of *Dalbergia* species, leaving the interspecific relationships within the genus largely unknown. The first comprehensive phylogenetic study of the genus *Dalbergia* was conducted by Vatanparast et al. (2013) [13] using ITS sequences and 64 species representing most of the recognized sections: i.e., sects. *Dalbergia* L.f., *Triptolemea* (Mart. ex Benth.) Benth, *Selenolobium* Benth, *Ecastaphyllum* (P. Browne) Ducke (Carvalho 1997), and *Dalbergaria* Prain. Their study confirmed the monophyly of a clade comprising *Dalbergia*, *Machaerium* and *Aeschynomene* sect. *Ochopodium*, but the relationships among them were unresolved. In addition, *Dalbergia* was resolved into five major clades, but many relationships among them and among species were poorly resolved. The previously recognized sects. *Triptolemea*, *Ecastaphyllum* and *Dalbergaria* were shown to be monophyletic, whereas sects. *Dalbergia* and *Selenolobium* were non-monophyletic. Therefore, more samples and molecular markers are needed to resolve the phylogenetic relationships among *Dalbergia*.

To date, there has been no formal biogeographical study of *Dalbergia* based on its phylogenetic scheme. Vatanparast et al. (2013) [13] hypothesized that *Dalbergia* originated in the New World and suggested that multiple migrations through LDD across oceans might account for its pantropical distribution. However, they did not conduct formal molecular dating or biogeographical analyses, so the origin date, original center, and the evolutionary history leading to the current disjunct distribution pattern of *Dalbergia* are still largely unknown. Therefore, formal dating and biogeological analyses should be conducted to infer the origin and account for the disjunct distribution pattern of the genus.

In this study, we used two plastid markers (*mat*K and *rbc*L) and one nuclear marker (ITS) representing 93 species of *Dalbergia* across its major lineages and entire geographical range, to reconstruct the phylogenetic relationships, estimate divergence times using multiple fossil calibrations and explore the original center and subsequent dispersal history of the genus *Dalbergia*.

## 2. Materials and Methods

### 2.1. Taxon Sampling and Sequence Alignment

Sequences of two chloroplast markers (*mat*K and *rbc*L) and one nuclear marker (ITS) representing 93 *Dalbergia* species and six outgroup species were downloaded from GenBank. Our samples represent the major clades and cover the entire geographic range distribution of *Dalbergia*. The outgroups include four species from the genus *Aeschynomene* L., one from *Machaerium* Pers., and one from *Pterocarpus* Jacq. If there was more than one sequence for the same marker of the species, we kept the longest. Details of samples used in this study are given in Table S1.

The *mat*K, *rbc*L and ITS loci were individually aligned using MAFFT v.7.4.0 [14] as implemented in Geneious v.8.1.9 [15] under default parameters. Alignments of the three loci were further individually inspected and manually adjusted in Geneious.

### 2.2. Phylogenetic Reconstruction of Dalbergia

In order to evaluate the conflicts among datasets, Incongruence Length Difference (ILD) testing [16] was performed in PAUP * v.4.0b.10 [17] between plastid (*rbc*L *and mat*K) and

nuclear regions (ITS), using a heuristic search with 1000 replicates, random taxa-addition, tree bisection and reconnection (TBR) branch-swapping saving 15 trees per replicate.

Prior to conducting the phylogenetic analyses, nucleotide substitution models were selected for each DNA region using the Akaike Information Criterion (AIC) implemented in jModelTest v.2.1.6 [18] (Table S2).

Phylogenetic analyses of the concatenated dataset were performed using Maximum Likelihood (ML) and Bayesian inference (BI). The ML analysis was performed using the IQ-Tree Web server [19]. We specified the best-fit model (GTR + I + G) selected by jModelTest and used the ultrafast bootstrap algorithm (UFBoot2) with 1000 replicates to assess branch support (-bb 1000), combined with a search of the best-scoring ML tree under default parameters. BI analyses were conducted using MrBayes v.3.2.6 [20]. We linked and specified GTR + I + G as the best-fit model according to the optimal scheme selected by jModelTest. Two independent Bayesian runs with four chains of Markov Chain Monte Carlo (MCMC) were run for 25 million generations, sampling every 10,000 generations. Chain convergence was checked in Tracer v.1.6 [21] by examining log likelihood plots and ensuring that Effective Sample Size (ESS) values were well above 200. After discarding 25% of the trees as burn-in, a majority rule consensus tree was constructed using TreeAnnotator v.2.3.2 [22]. Outputs of all phylogenetic analysis were read using FigTree v.1.4.2 [23] and nodes with ultrafast bootstrap (BS) > 95% [24] and posterior probability (PP) $\geq$ 0.90 [20] were considered well supported. We also conducted phylogenetic analyses of ML and BI for each DNA region using the best-fit model selected by jModelTest (Table S2). Other parameters were set similarly.

*2.3. Divergence Time Estimation*

Divergence time estimations were generated on the BI tree of the concatenated dataset using BEAST v.1.8.4 [22], which was modeled under a Yule process using a random starting tree and an uncorrelated relaxed clock. Within BEAST, a Birth-Death model was employed for tree priors, the GTR + G + I evolution model with four gamma categories was applied based on AIC results from jModelTest, and other parameters were set as default values. Four calibration points were used, with the root node calibrated at the maximum age of 96.33 Ma (Legumes stem) based on the meta-calibration study of flowering plants by Magallón et al. (2015) [25], as shown in Table 1. The prior distributions of the root node and the other calibration points were set as log normal with a standard deviation of 1.0.

**Table 1.** Fossils used as calibration points to generate a time-calibrated phylogenetic tree of *Dalbergia*. All ages in millions of years (Ma) and set to a minimum, except for Label A which was set to a maximum.

| Label | Node Constrained (MRCA) | Species | Morphology | Age | References |
|---|---|---|---|---|---|
| A | Legume stem | *Polygala californica–Cercis occidentalis* | | 96.33 | Magallon et al. (2015) |
| B | *Styphnolobium* stem | *Styphnolobium japonicum–Pickeringia montana* | Leaf and fruit | 40 | Lavin et al. (2005) |
| C | *Tipuana* stem | *Tipuana tipu–Pterocarpus indicus* | Fruit | 10 | Lavin et al. (2005) |
| D | *Dalbergia* stem | *Dalbergia hupeana–Machaerium lunatum* | Leaf | 40 | Lavin et al. (2005) |

The BEAST file was generated in BEAUti v.1.8.4 [22]. Then, we conducted two runs of four Markov chains for 40 million generations with sampling every 1000 generations. The output files were examined in Tracer to evaluate convergence of the runs and the ESS ($\geq$200)

for all parameters. The runs were combined using LogCombiner v.1.8.2 [22]. Following the removal of the first 20% samples as burn-in, the sampled posterior trees were summarized using TreeAnnotator v.1.8.4 [22] to generate a maximum clade credibility (MCC) tree and calculate the mean ages, 95% highest posterior density intervals (95% HPD), and PP. The chronogram was visualized and annotated using FigTree.

*2.4. Ancestral Area Estimations*

Ancestral area estimations were conducted with the maximum likelihood framework using the package BioGeoBEARS v.1.1.2 [26] in RASP v.4.2 [27]. The analysis was carried out on the chronogram tree from the BEAST analysis without the outgroups. Three models were tested, including Dispersal–Extinction–Cladogenesis (DEC; [28]), Dispersal Vicariance Analysis like (DIVALIKE; [29]), and Bayesian Analysis like (BAYAREALIKE; [30]). Given the ongoing debate around the use of founder-event speciation + j, we did not implement this parameter in our reconstructions [31]. Using the model selection function, the best-fit model was selected by comparing the AIC criterion among all models (Table S3). The maximum areas parameter was set according to the maximum number of areas occupied by any extant species in the dataset. Consequently, a maximum of three areas was chosen.

Five biogeographic areas were defined, based on the respective distribution of the species: A—Australasia; B—Africa; C—Madagascar; D—Central America (including Florida and Caribbean); E—South America. We compiled the information about species distribution from the literature [2,32–35], herbarium specimens and online databases (www.gbif.org, accessed on 2 May 2021; www.plantsoftheworldonline.org, accessed on 3 March 2021).

## 3. Results

### 3.1. Phylogenetic Analyses

The ILD test showed no significant incongruence between nuclear and plastid datasets ($p = 0.37$). Alignment length, numbers of variable sites, and parsimony informative sites for each marker and the concatenated dataset are provided in Table 2.

**Table 2.** Features of the DNA data sets used in this study (bp = base pairs).

| DNA Region | Alignment Length (bp) | Number of Variable Sites | | Number of Potentially Informative Sites | |
|---|---|---|---|---|---|
| *rbc*L | 695 | 17 | (2.44%) | 24 | (3.45%) |
| *mat*K | 1070 | 74 | (6.91%) | 119 | (11.12%) |
| ITS | 1041 | 105 | (10.08%) | 338 | (32.46%) |
| Concatenated dataset | 2806 | 196 | (6.98%) | 481 | (17.14%) |

Topology of the BI tree was largely congruent with the ML tree described (Figures 1 and S1). Thus, we mapped support values of the BI tree onto the ML tree (Figure 1). The genus *Dalbergia* was strongly supported as monophyletic (BS = 100, PP = 1.00) and five major clades were recovered, labelled as clades I–V (Figure 1). All clades were strongly supported by all our analyses, except clade IV which was moderately supported in the ML analysis with BS = 89 (Figure 1). Clade I comprising two South American species (*D. miscolobium* Benth. and *D. spruceana* Benth.) was resolved as sister to all other *Dalbergia* species with strong support (BS = 98, PP = 1.00). Clade II comprising eight South American species was resolved as the second divergent clade with strong support (BS = 99, PP = 1.00). Clades I and II include species from sects. *Dalbergia* and *Selenolobium*, and all species of these two clades are mainly from the Neotropical regions. Clade III includes three well-supported subclades (III-a, III-b and III-c). Subclade III-a, comprising three Afro-American species (seven accessions) with the African *D. adamii* Berhaut being sister to the Neotropical sect. *Ecastaphyllum*, was resolved as sister to a clade comprising subclade III-b and subclade III-c. All species in subclades III-b and III-c are from the Australasian region (Figure 1). Clade IV is composed of two highly sup-

ported subclades (IV-a and IV-b). Within subclade IV-a, an African–Malagasy clade (M1) was recovered as sister to an entire Australasian clade. Within the subclade IV-b, an African–Malagasy clade (M2) was supported as sister to sect. *Dalbergaria* from the Australasian region. Similarly, Clade V was strongly supported (BS = 98, PP = 1.00) and split into four subclades (V-a, V-b, V-c, and V-d). Within subclade V-a, *D. nigra* (Vell.) Allemao ex Benth. from South America were resolved as sister to a clade comprising two species from Australasia (*D. sandakanensis* Surnamo & H.Ohashi and *D. bintuluensis* Surnamo & H.Ohashi) and two from Africa (*D. armata* E.Mey. and *D. hostilis* Benth.). Subclade V-b comprised two species from Australasia and two from Africa, and was supported as sister to the remaining species in Clade V with low support (BS = 67, PP = 0.57). All species in Subclade V-c are from Australasia, but monophyly of Subclade V-c was weakly supported. Finally, within subclade V-d, *D. bracteolata* Baker from Africa and Madagascar was supported as sister to the remaining species of this subclade, only in the ML tree (BS = 95). In addition, subclade V-d includes the monophyletic sect. *Triptolemea* from South America, the African-Malagasy clade (M3) and some dispersive lineages from Central America, Africa and Australasia.

*3.2. Divergence Times and Biogeographical Analyses*

The stem age of *Dalbergia* was inferred to be *c*. 40.7 Ma (95% HPD: 41.2–40.2 Ma) and the crown age to be *c*. 22.9 Ma (95% HPD: 25.9–19.9 Ma; nodes 1 and 2 respectively, Figure 2). Diversification of the main clades occurred from the Early to Late Miocene (nodes 3, 4, 5, 7 and 9, Figure 2)—i.e., from *c*. 19.0 Ma (95% HPD: 21.7–16.3 Ma) to *c*. 6.7 Ma (95% HPD: 9.5–4.5 Ma). Clades I and II (species from sects *Dalbergia* and *Selenolobium*) diverged during the Late Miocene *c*. 6.7 Ma (95% HPD: 9.5–4.5 Ma; node 3, Figure 2) and the Middle Miocene *c*. 15.9 Ma (95% HPD: 19.4–12.6 Ma; node 4; Figure 2), respectively. The divergence of sect. *Ecastaphyllum* was dated in the Pleistocene around 2.2 Ma (95% HPD: 3.4–1.2 Ma; node 6, Figure 2). Sect. *Dalbergaria* diverged in the Middle Miocene around 14.2 Ma (95% HPD: 16.6–12.1 Ma; node 8, Figure 2) and sect. *Triptolemea* was dated to the Pliocene *c*. 3.3 Ma (95% HPD: 4.7–2.1 Ma; node 10, Figure 2).

Ancestral range estimation recovered the DEC model as the best-fit model (lnL = −150.4 and AICwt = 1; Table S3). This result showed that long distance dispersal may have played an important role in the biogeographical history of *Dalbergia*, as shown by the values obtained for the two parameters of the analysis (d = 0.0063 and e = 0.0028, Table S3).

The ancestral area reconstruction showed that South America was the ancestral area of *Dalbergia* (node 1, Figure 3; Table 3) with high probability (*p* = 100%). Subsequently, it expanded into other regions through dispersal (Figures 3 and 4). South America was also inferred as the ancestral range for both Clades I and II (node 2 and 3, Figure 3; Table 3) with *p* = 100% for the two nodes. The DEC model inferred an Australasia–South America origin for Clade III (node 4, Figure 3; Table 3). Clades IV and V were estimated to be from Australasia (nodes 9 and 13, Figure 3; Table 3) with moderate probability *p* = 75% and *p* = 63%, respectively.

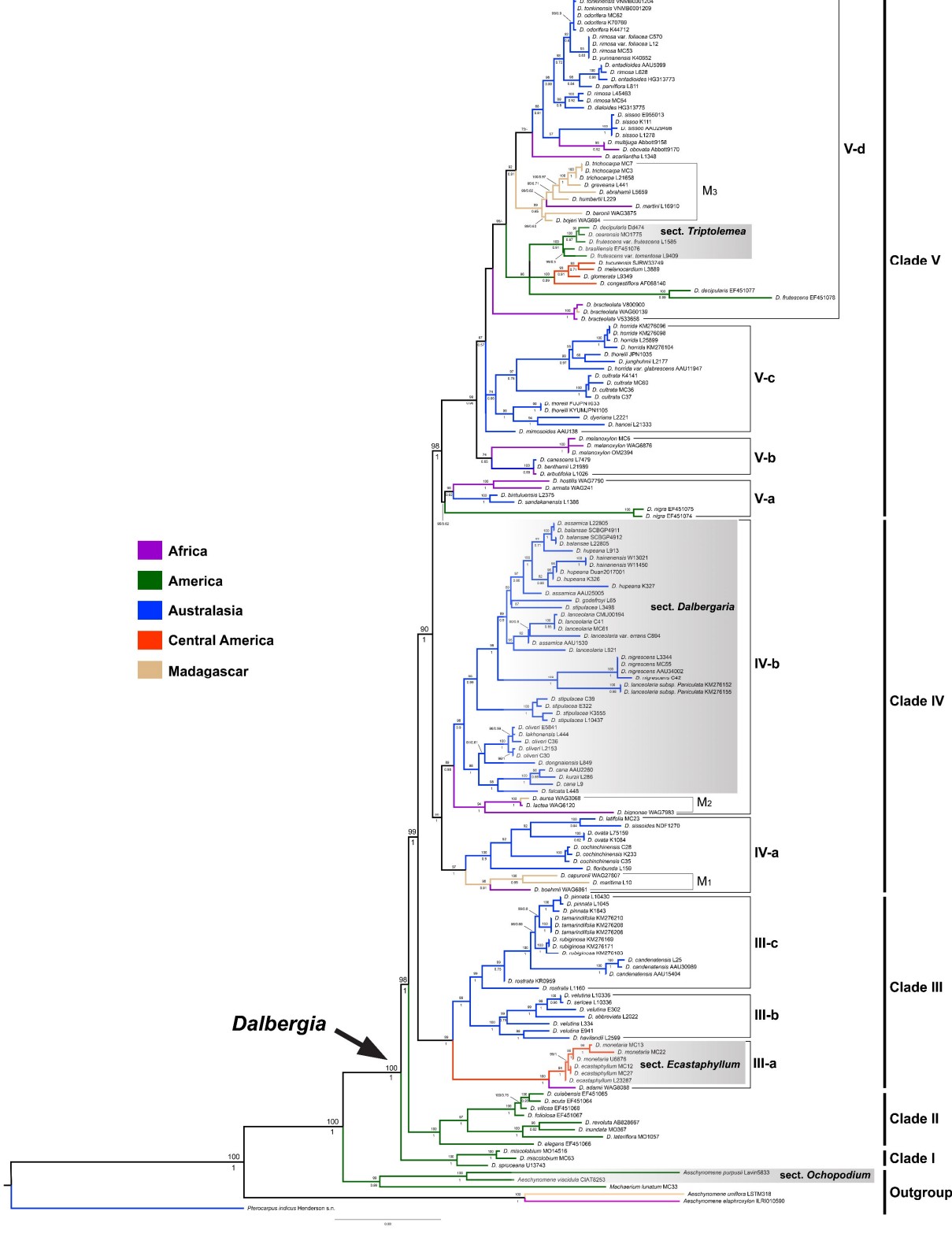

**Figure 1.** Majority rule consensus tree inferred from the ML analysis based on the concatenated data matrix (*mat*K, *rbc*L and ITS) showing the relationships among 93 species of *Dalbergia* and 6 outgroups. M1, M2, and M3 represents the Madagascar clades. Numbers along branches are ML bootstrap values (up) and BI posterior probabilities (down), respectively. A dash means the topology is not supported by the BI tree. Values <50% and <0.5 are not shown.

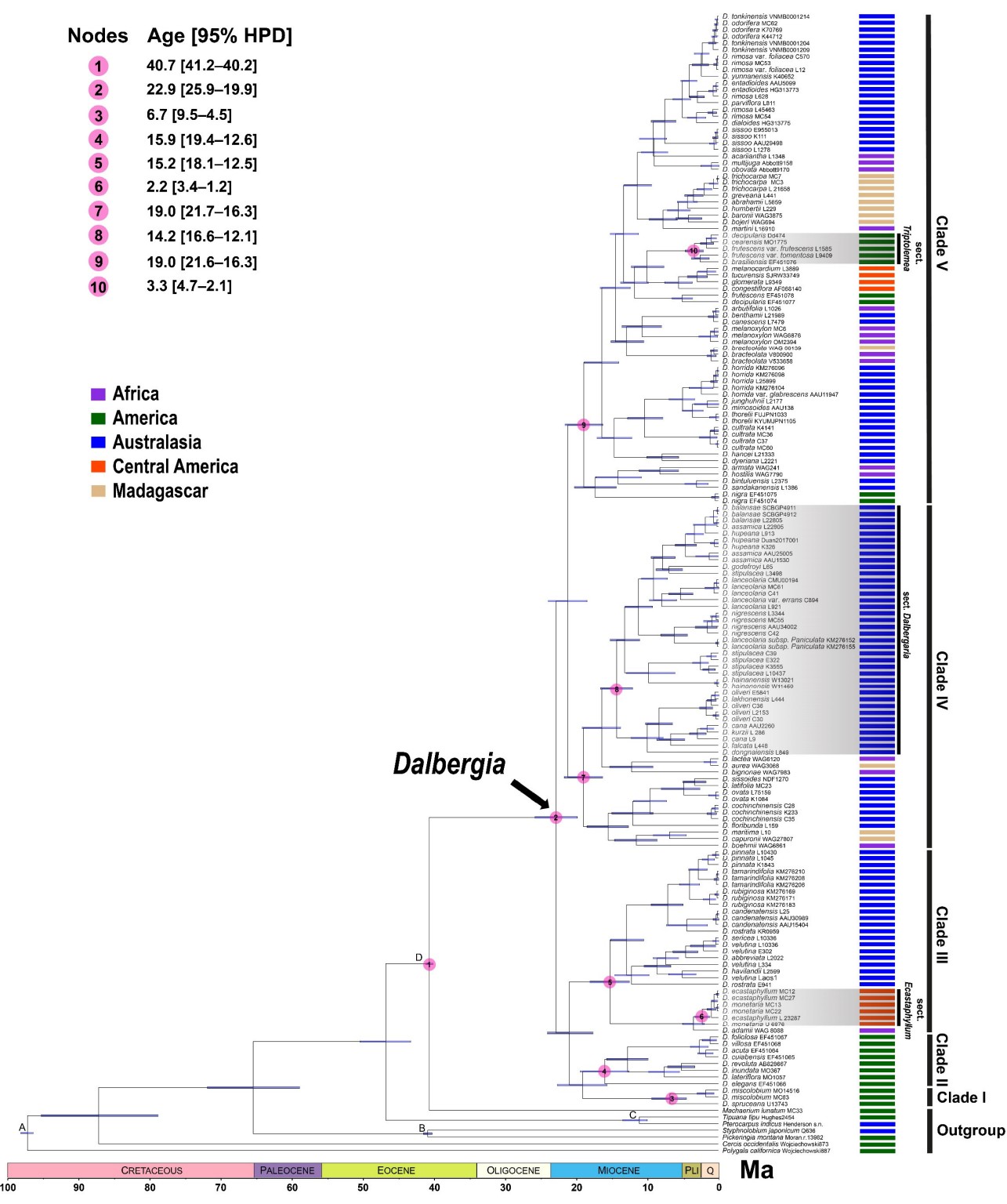

**Figure 2.** Chronogram of *Dalbergia* inferred from the BI tree using BEAST. The shaded blue horizontal bars show 95% HPD for the divergence times. Labels A–D correspond to the calibration points used (see Table 1 for details). Nodes of interests are marked as 1–10.

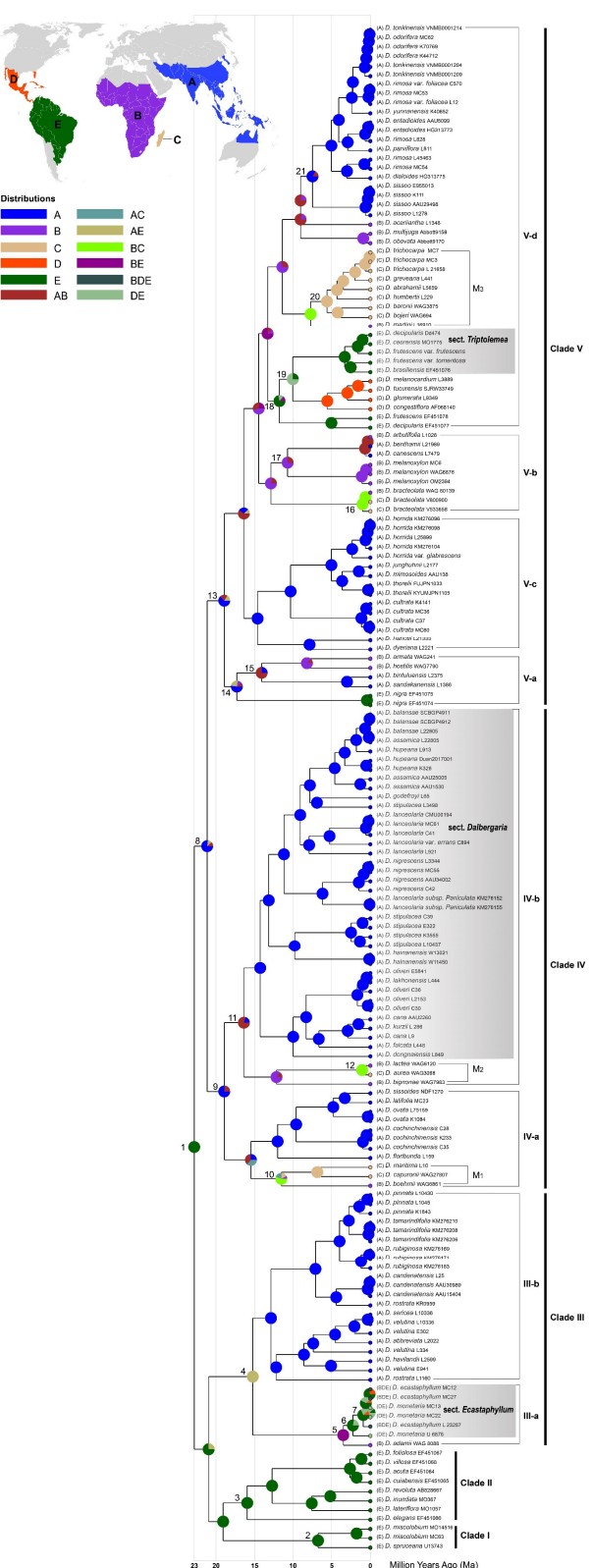

**Figure 3.** Ancestral area reconstructions under the dispersal–extinction cladogenesis (DEC) model in the BioGeoBEARS package as implemented in RASP; the relative probabilities of alternative ancestral areas are shown by pie charts at each node. Nodes of interests are marked 1–21. Area abbreviations are as follows: A—Australasia; B—Africa; C—Madagascar; D—Central America (including Florida and the Caribbean); E—South America. M1, M2, and M3 represent the Madagascar clades.

**Table 3.** Results of divergence time and ancestral area estimation for some major nodes in *Dalbergia*. The 95% highest posterior density (HPD) values of the divergence time estimations are provided in square brackets. The estimated ancestral areas of each node as inferred by BioGeoBEARS using the DEC model are shown, with rounded probabilities given as percentages. When only one area is shown, the probability is 100%. Areas are abbreviated as follows: A—Australasia; B—Africa; C—Madagascar; D—Central America (including Florida and the Caribbean); E—South America. Node numbers refer to Figure 3.

| Node | Estimated Divergence Time in Ma with [95% HPD] | Estimated Ancestral Area (DEC) |
|------|-----------------------------------------------|--------------------------------|
| 1 | 22.9 [25.9–19.9] | E |
| 2 | 6.7 [9.5–4.5] | E |
| 3 | 15.9 [19.4–12.6] | E |
| 4 | 15.2 [18.1–12.5] | AE |
| 5 | 3.4 [5.1–2.0] | B |
| 6 | 2.2 [3.4–1.2] | E 78; DE 22 |
| 7 | 0.9 [1.7–0.3] | E 68; DE 21; D 11 |
| 8 | 21.1 [24.0–18.5] | A 77; AE 12; AB 11 |
| 9 | 18.9 [21.7–16.3] | A 75; AB 25 |
| 10 | 11.5 [14.6–8.6] | BC 55; AC 20; C 17; B 8 |
| 11 | 16.3 [19.2–13.8] | AB 79; A 21.62 |
| 12 | 1.0 [2.1–0.3] | BC |
| 13 | 18.9 [21.6–16.3] | A 63; AB 20; AE 17 |
| 14 | 17.3 [20.3–14.4] | A 52; AE 26; B 11; AB 11 |
| 15 | 14.1 [17.3–10.9] | AB 76; A 24 |
| 16 | 1.0 [1.7–0.4] | BC |
| 17 | 10.7 [13.6–8.0] | B 75 AB 25 |
| 18 | 11.7 [13.8–9.0] | E 72; DE 15; BE 14 |
| 19 | 10.0 [12.3–7.7] | DE 74; E 26 |
| 20 | 5.6 [7.3–4] | C |
| 21 | 7.5 [9.4–5.9] | A 84; AB 16 |

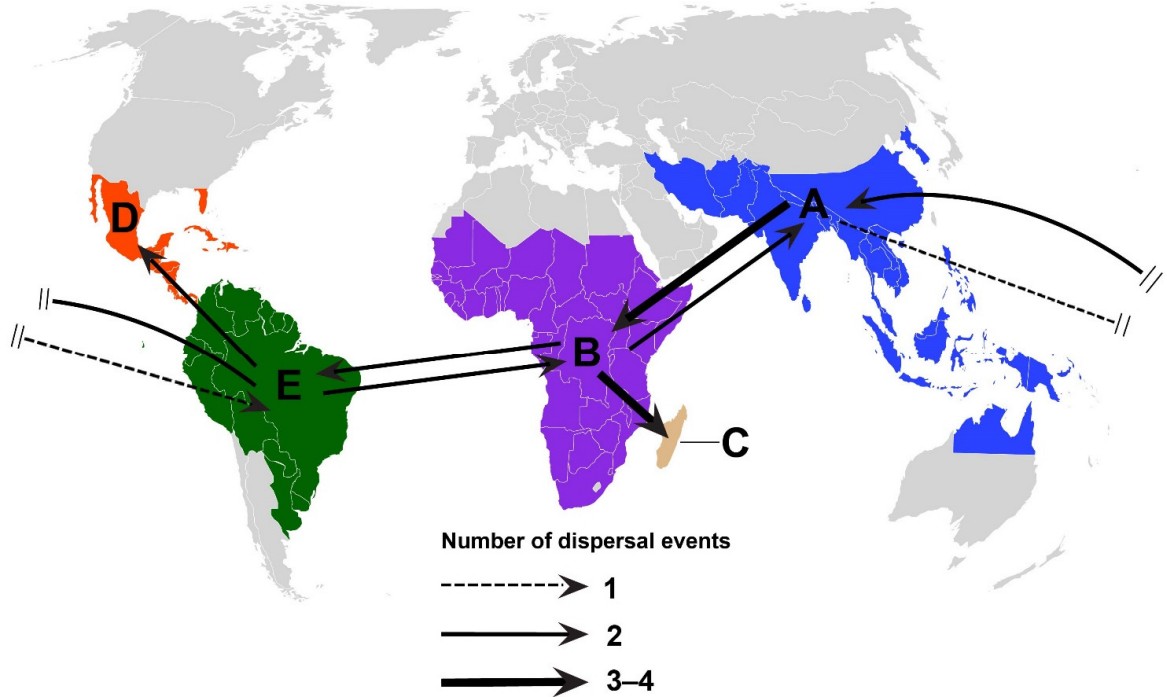

**Figure 4.** Major dispersal/migration routes suggested by DEC analyses of *Dalbergia*. Arrows indicate the dispersal direction; line thickness is proportional to the number of events. Labels A, B, C, D and E represent Australasia, Africa, Madagascar, Central America (including Florida and the Caribbean) and South America, respectively.

## 4. Discussion

### 4.1. Phylogenetic Relationships and Taxonomic Implications

Our study confirmed the monophyly of *Dalbergia* and resolved the genus to be sister to a clade comprising *Aeschynomene* sect. *Ochopodium* and *Machaerium*, as suggested in previous studies which focused on the relationships between *Dalbergia* and relatives [11,12]. Our analysis resolved the genera into five major clades, which is largely congruent with previously published phylogeny using nuclear DNA region ITS of 64 *Dalbergia* species [13]. However, our results strongly supported each clade and their relationships, except for clade IV in the ML tree (BS = 89, Figure 1), whereas these supports were weak in previous studies [13]. In addition, resolution and support for the vast majority of the nodes in our study have been improved compared to previous studies, e.g., Vatanparast et al. (2013) [13] and Ribeiro et al. (2007) [12]. This is attributed to our increased selection of samples and markers compared to previous studies, which were based on limited samples [11,12] or only one marker [13].

Within the five previously defined sections, four are from the Neotropics and only one from Australasia. Among those of the Neotropics, sect. *Ecastaphyllum* characterized by racemose or paniculate inflorescences and orbicular to suborbicular or reniform fruits, and sect. *Triptolemea* with cymose inflorescences and thin samaroid pods, were seen in our analyses to be monophyletic, as previously suggested in studies using both morphological and molecular data [12,13,36]. However, sect. *Dalbergia,* characterized by paniculate inflorescences and samaroid pods, and sect. *Selonolobium*, with racemose or paniculate inflorescences and crescent or kidney-shaped fruits, are non-monophyletic. Species of these two sections resolved into two diverged clades (Clade I and Clade II; Figures 1 and S1), each clade comprising species from each of these two sections. One exception is *D. nigra* from sect. *Dalbergia*, which has distinct phylogenetic position in Clade V according to our analyses. This species also has very distinct traits including a calyx with a glabrous tube and pilose teeth, obovate standard petals, and dark brown glossy fruits lacking prominent venation, as reported by Carvalho (1989) [36]. Among the species from Australasia, our results strongly supported sect. *Dalbergaria* to be monophyletic, whereas this support value was low in a previous study [13]. This section has Southeast Asian distribution and is characterized by reflexed standard petals and stamens that are usually in two bundles of five. In our study, many species were not included within any of the five above-mentioned sections. Thus, a through revision of the intrageneric *Dalbergia* classification integrating morphological traits and denser sampling phylogeny is urgently needed.

### 4.2. Origin and Biogeographical Diversification of Dalbergia

South America was inferred in our study to be the origin center of *Dalbergia* (Figure 3; Table 3), which is slightly different from the hypothesis proposed by Vatanparast et al. (2013) [13], who suggested a Neotropical (Central and South America) origin for this genus. This is probably due to differences in sampling and the definition of biogeographic regions. Despite these differences, South America was identified as an important region for the origin of *Dalbergia*. Our estimated stem age of *Dalbergia* (*c*. 40.7 Ma) is largely consistent with the *c*. 40.4–43.0 Ma estimated by Lavin et al. (2005) [37] which was based on the whole legumes family. However, our estimated crown age (*c*. 22.9 Ma; Early Miocene; see node 2, Figure 2) for *Dalbergia* is much older than *c*. 14.7 Ma (Middle Miocene) inferred by Hung et al. (2020) [38], who only sampled six representative species from the most basal Clade I and other clades of *Dalbergia*. Additionally, the use of different fossil calibrations probably contributed to the difference in the estimation of crown age between the two studies. Multiple fossil species have been reported; *Dalbergia phleboptera* is the earliest detected *Dalbergia* fossil species dated to 23.0–27.8 Ma (Late Oligocene) from France [39], then 15.97–23.03 Ma (Early Miocene) fossil species *D. nostratum* from Slovakia [40], and the later Miocene fossil (5.33–11.61 Ma) from China [41]. However, these fossils could not be

confidently placed within the genus because of their limited morphological traits, thus they were not used time dating in our study.

Vicariance has been widely applied to explain the intercontinental distributions of biological taxa across broad oceans [42], but a series of molecular phylogenetic and biogeographical studies suggested that LDD should account for most of these disjunctions, mainly because the estimated divergence times are much younger than the continental breakup times inferred from geological data [43]. Thus, the finding that *Dalbergia* arose during the Early Miocene allows us to dismiss the hypotheses based on vicariance, and to take into account LDD as the most possible explanation for the pantropical pattern disjunct distribution of this genus. In addition, some previous legumes studies showed that LDD played an important role in the biogeography of several legume taxa, including *Apios* [44], *Canavalia* [45], *Zornia* [46], the tribe Fabeae [47] and legumes in general [48,49].

Our analysis suggested at least seven migrations between the Neotropics (South America) and the Paleotropics. Four of these were inferred to be from the Neotropics to the Paleotropics and took place during the Early Miocene to the Pleistocene (*c.* 21.6–0.9 Ma; nodes 4, 5, 7 and 8, Figures 3 and 4; Table 3), while three migrations from the Paleotropics to the Neotropics were inferred and were dated from the Early Miocene to the Pleistocene (*c.* 17.3–2.2 Ma; nodes 6, 14 and 18, Figure 3; Table 3). The estimated ages of these migrations are too young to support any hypothesis involving continental drift and early Tertiary biotic interchange. This suggests that the most probable scenario which to explain these tropical disjunctions involved transoceanic LDD by ocean currents. Generally, *Dalbergia* species present samaroid pods and buoyant seeds which are adapted to water dispersal [2,50].

Among the Neotropics, at least two dispersal events between South America and Central America have been inferred. One migration from South America to Central America was dated in the late Miocene at *c.* 10.0 Ma (nodes 19, Figures 3 and 4; Table 3). Although the emergence time of the Isthmus of Panama remains contentious, the authors assumed that the latest date for closure of this landmass which connects South and Central America was around 3.5 Ma [51–53]. Therefore, this first migration was probably through transoceanic LDD via ocean currents, birds, or wind. This is consistent with a wave of plant dispersal between South and Central America predating the closure of the Isthmus [51,54]. In addition, several examples of putatively pre-Isthmus dispersal from South America to Central and North America have been documented [55–58]. The second migration took place in the Early Pleistocene at *c.* 2.2 Ma (nodes 6 and 7, Figure 3; Table 3), after the closure of the Isthmus of Panama, and led to the distribution of sect. *Ecastaphyllum* across South and Central America. Thus, this dispersion probably followed the Isthmus which established an important route of migration between South, Central and North America, giving rise to the Great American Biotic Interchange [43,59].

At least two dispersal events from Africa to Australasia and four reverse dispersals from Australasia to Africa in the Early to Late Miocene were inferred in our analyses (nodes 9, 11, 13, 14, 15 and 21, Figures 3 and 4; Table 3). Boreotropical migration [60], rafting of the Indian subcontinent [61], transoceanic dispersal [62,63], or Miocene overland migration across the Arabian Peninsula [64–66] have all been invoked to explain intercontinental dispersal between Africa and Australasia. The young age of the migrations between Africa and Australasia in our analyses favors Miocene overland migration or transoceanic LDD. During the Early to the Middle Miocene, a land connection was formed between Africa and Southwest Asia which corresponded with a global warming phase that peaked from 17 to 15 Ma [67–69]. This probably played an important role in these migrations, and several studies have shown similar patterns of dispersal [43,66,70]. However, many *Dalbergia* species are adapted to water dispersal [2] and have winged fruits [71]. Therefore, transoceanic LDD between Africa and Australasia cannot be ruled out.

The Malagasy *Dalbergia* were resolved in three clades (M1, M2 and M3), nested with other species from Africa (Figures 1 and 3). This occurrence in three distinct clades and the inference of an African species as sister to each Malagasy clade suggests at least three independent migrations from Africa to Madagascar during the Late Miocene to the

Pleistocene (*c*. 11.5–1.0 Ma; nodes 10, 12 and 20, Figures 3 and 4; Table 3). Given these dates, it is evident that these splits occurred after Madagascar separated from Africa [72]. Thus, both migrations from continental Africa to Madagascar were probably achieved through LDD across the Mozambique channel, which supports the predominant biogeographical pattern found by Yoder and Nowak (2006) [73]. In addition, LDD from continental Africa to Madagascar has been suggested for many taxa of animals and plants [63,74–77].

## 5. Conclusions

Although our study corroborates previous findings (e.g., the monophyly of *Dalbergia* and the presence of five major clades), it sheds new light on the relationships among the major clades of the genus. These were resolved in our phylogenetic analyses with strong support, and an increase of support for the vast majority of the nodes (around 85% of the nodes). In addition, we have provided the first detailed divergence times and biogeographical history study of the genus. We inferred a Middle Eocene and Early Miocene origin for the stem and crown of *Dalbergia*, respectively. The genus originated in South America and achieved its present-day pantropical distribution largely through recent transoceanic long-distance dispersal. However, taxon sampling included in this study included 39% of the total species diversity of the genus. Therefore, we recommend future studies to harness more loci and increased taxonomic sampling for deeper resolution of the phylogeny of the genus *Dalbergia*.

**Supplementary Materials:** The following supporting information can be downloaded at: https://www.mdpi.com/article/10.3390/agronomy12071612/s1, Figure S1: Bayesian 50% consensus tree of *Dalbergia* resulting from the combined nuclear (ITS) and plastid (*mat*K and *rbc*L) datasets. Bayesian posterior probability (only values >0.50) are presented above the branches; Table S1: Species list, voucher information and GenBank data accession number of the taxa used for the analysis; Table S2: Nucleotide substitution models selected using jModelTest under the Aikake Information Criterion (AIC) for each DNA region; Table S3: Likelihood (LnL) and Akaike Information Criterion (AIC) scores from each of the models tested in BioGeoBEARS implemented in RASP v.4.2 for the ancestral area estimations analyses. The best model is highlighted in bold.

**Author Contributions:** F.R.R. and T.-S.Y. designed the research; F.R.R., O.O. and R.Z. performed the research and analyzed the data; F.R.R., G.W.S. and T.-S.Y. wrote the first version of the manuscript which was reviewed and agreed by all co-authors. All authors have read and agreed to the published version of the manuscript.

**Funding:** This research was supported by the National Natural Science Foundation of China, key international (regional) cooperative research project (No. 31720103903), the Strategic Priority Research Program of Chinese Academy of Sciences (XDB31000000), the Science and Technology Basic Resources Investigation Program of China (2019FY100900), the Large-scale Scientific Facilities of the Chinese Academy of Sciences (No. 2017-LSF-GBOWS-02), and the National Natural Science Foundation of China (Project No. 31270274), the Yunling International High-end Experts Program of Yunnan Province, China (grant No. YNQR-GDWG-2017-002 to P.S.S. and T.-S.Y. and YNQR-GDWG-2018-012 to D.E.S. and T.-S.Y.), China Postdoctoral Science Foundation (2020M683391) and Postdoctoral Directional Training Foundation of Yunnan Province.

**Informed Consent Statement:** Not applicable.

**Data Availability Statement:** Not applicable.

**Acknowledgments:** We are grateful to the Germplasm Bank of Wild Species in Kunming Institute of Botany (KIB), including the iFlora high performance computing center, for facilitating this study. We thank Maria Vorontsova from Royal Botanical Gardens Kew, Richmond, UK for the review of the previous version of the manuscript.

**Conflicts of Interest:** The authors declare no conflict of interest.

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
