# Peer review of "A Dated Phylogeny of the Pantropical Genus Dalbergia L.f. (Leguminosae: Papilionoideae) and Its Implications for Historical Biogeography"

_agronomy, doi:10.3390/agronomy12071612_

Round 1

Reviewer 1 Report

The article entitled “A dated Phylogeny of the pantropical genus Dalbergia L.f. (Leguminosae: Papilionoideae) and its implications for historical biogeography.” In this study, the author used two plastid DNA markers (matK and rbcL) and one nuclear marker (ITS) of 93 Dalbergia species from across the genus range. Its main lineages were used in phylogenetic analysis. The overall study covered a very important research topic.

The abstract need to be revised. Currently, the structure of the abstract is not according to the scientific article.

The introduction is well written but needs more literature to be cited, i.e., DOI: 10.11646/phytotaxa.538.3.2    DOI: 10.36253/caryologia-1056

The methodology of the article is enough for this study. The methodology is modern and acceptable for this work, but the marker used maybe not be enough to differentiate the species of the genus Delbergia. So, that is why some of the species fall in the same clade.

In the ML analysis, some species boundaries are hard to differentiate? Do the authors have any evidence of morphological/ecological characteristics to differentiate such species?

The text is not constant in the overall manuscript. Some text font is different. See the discussion and conclusion portion of the article.

The discussion is well written, but the comparative results of the previous studies must be included in more detail.

Reviewer 2 Report

Overview of the manuscript:

There has clearly been a lot of work put into this manuscript making the evolutionary diversification of the genus to be clear, despite the lack of new molecular data. There are some very nice results and the figures are particularly good.

Need to revise the font and police throughout

Abstract: 
lines 13-14: Please rearrange these lines …
many relationships among the major clades and species are still unresolved. In addition, where this genus originated and how the current disjunct distribution pattern formed have been largely unaddressed

line 22: achieved its current pantropical

Introduction:

line 29 remove moist and dry ; to read:”Dalbergia species…. Including tropical forests, savannas,…

Line 30: The Dalbergia species display a high diversity of life forms, including trees, multibranched shrubs, and woody lianas

line 32: Please list some famous Madagascar rosewood 

line 38: remove “still very”

lines 38-40: suggestion

The intrageneric classification of Dalbergia is limited. Since the description of the genus in XXX, only Bentham (1860) has carried out large taxonomic scales of genus and  reported the currently known 64 Dalbergia species, which were divided into six XXX

Lines 40-41: Later, this treatment was followed by subsequent regional monographs based on morphological characters alone.

Materials and Methods

Line 94: in PAUP* v.4b10

Lines 97-100: repetition of mentioned below

Line 106: What are the settings of searches for partition scheme? Please add

-->ML and BI analyses were performed using XXX and XXXX models, respectively.

Lines 115: In ultrafast bootstrap of IQTREE, only when BS > 95% should the clade be believed

Table 1

Age (Ma)

Lines 140-141: …were conducted with the maximum likelihood framework described using the package

Lines 142-146: Suggestion: Three models were tested: Dispersal-extinction-cladogenesis (DEC; Reference), dispersal-vicariance-like (DIVALIKE; Reference), and BAYAREALIKE (Bayesian analysis; Reference). Given the ongoing debate around the use of founder-event speciation +j, we did not implement this parameter in our reconstructions (see Ree and Sanmartín, 2018? list the appropriate reference). Model choice was based on the Akaike Information Criterion (AIC) values (reference)

Results

Line 157: Please add the meaning ILD. To read: The incongruence length difference test (ILD)…

Line 161: were strongly recovered as : The monophyly of the genus Dalbergia was strongly supported ((BS = 100, PP = 1; Fig. 1) and five major clades were recovered, labeled as clade I –IV

 as

Line 162: 1.00 not 1 (please change throughout the MS)

Line 169: includes

Line 174: are mainly comprised of: “….are mainly composed by the Australasian species”

Line 176: is composed: “… included two highly...”

Fig. 1: Perhaps you can use dash // and reduce the length of the 3 basal outgroup. Also Not sure why you have used these polyphyletic outgroup species, is it because of the calibration in Fig. 2? The bootstrap supports might have been higher by excluding the long branch species P. indicus  

Line 199: The genus Dalbergia originated in the Middle Eocene and the early Miocene: this range period does not meet with the 95% HPD (height posterior density).

Line 200: The crown age of Dalbergia was estimated to be in the Miocene (22.9 Ma (95% HPD: 25.9 – 19.9 Ma; nodes 1 and 2 respectively, Figure 2).

Discussion

Line 245 monophy: monophyly

Line 344-345: Given these dates, it’s evident that these splits occurred after Madagascar rifted from Africa

5. Conclusions

L 356-358. Please highlight your main findings following the objectives of this paper. 

The manuscript is not always well written at least in some parts. Unfortunately, there are several grammatical, academic, and typographical mistakes in the text that need to be fixed. The text given in the introduction and discussion sections should be checked carefully by the authors. Prior to re-submission, I suggest submitting the manuscript to a native English speaker.

Round 2

Reviewer 1 Report

Thanks to all the authors for incorporating all the suggested changes. I believe the manuscript has been sufficiently improved for publication in Agronomy.

TRANSLATE with x English
Arabic Hebrew Polish
Bulgarian Hindi Portuguese
Catalan Hmong Daw Romanian
Chinese Simplified Hungarian Russian
Chinese Traditional Indonesian Slovak
Czech Italian Slovenian
Danish Japanese Spanish
Dutch Klingon Swedish
English Korean Thai
Estonian Latvian Turkish
Finnish Lithuanian Ukrainian
French Malay Urdu
German Maltese Vietnamese
Greek Norwegian Welsh
Haitian Creole Persian  
TRANSLATE with COPY THE URL BELOW Back EMBED THE SNIPPET BELOW IN YOUR SITE Enable collaborative features and customize widget: Bing Webmaster Portal Back

Reviewer 2 Report

Corrections made by the authors are satisfied.